# Genetically Engineered Filamentous Bacteriophages Displaying TGF-β1 Promote Angiogenesis in 3D Microenvironments

**DOI:** 10.3390/jfb15110314

**Published:** 2024-10-24

**Authors:** In-Hyuk Baek, Volkhard Helms, Youngjun Kim

**Affiliations:** 1Environmental Safety Group, Korea Institute of Science & Technology Europe GmbH, Campus E71, 66123 Saarbrücken, Germany; baekja85@gmail.com; 2Center for Bioinformatics, Saarland University, 66123 Saarbrücken, Germany; volkhard.helms@bioinformatik.uni-saarland.de

**Keywords:** filamentous bacteriophage, angiogenesis, HUVECs, bacteriophage display, lab-on-a-chip, ECM

## Abstract

Combined 3D cell culture in vitro assays with microenvironment-mimicking systems are effective for cell-based screening tests of drug and chemical toxicity. Filamentous bacteriophages have diverse applications in material science, drug delivery, tissue engineering, energy, and biosensor development. Specifically, genetically modified bacteriophages have the potential to deliver therapeutic molecules or genes to targeted tumor tissues. The engineered bacteriophages in this study significantly enhanced endothelial cell migration and tube formation within the extracellular matrix (ECM). Compared to TGF-β1 alone and non-modified phages, the presence of TGF-β1 on the bacteriophages demonstrated superior performance as a continuous stimulant in the microenvironment, effectively promoting these angiogenic processes. Assays, including RT-qPCR, ELISA, and fluorescence microscopy, confirmed the expression of angiogenic markers such as CD31, validating the formation of 3D angiogenic structures. Our findings indicate that the TGF-β1 displayed by bacteriophages likely acted as a chemotactic factor, promoting the migration, proliferation, and tube formation of endothelial cells (ECs) within the ECM. Although direct contact between ECs and bacteriophages was not explicitly confirmed, the observed effects strongly suggest that TGF-β1-RGD bacteriophages contributed to the stimulation of angiogenic processes. The formation of angiogenic structures by ECs in the ECM was confirmed as three-dimensional and regulated by the surface treatment of microfluidic channels. These results suggest that biocompatible TGF-β1-displaying bacteriophages could continuously stimulate the microenvironment in vitro for angiogenesis models. Furthermore, we demonstrated that these functionalized bacteriophages have the potential to be utilized as versatile biomaterials in the field of biomedical engineering. Similar strategies could be applied to develop angiogenic matrices for tissue engineering in in vitro assays.

## 1. Introduction

Vascular systems ensure a constant circulation of oxygen, nutrients, blood, and lymphocytes as the basis for the survival and homeostasis of the body [1]. They are composed of an intricate network of capillaries, veins, and vessels, all lined with a single layer of squamous endothelial cells (ECs).

These ECs exhibit a thick layer in a heterogeneous mixture of vessels [2]. Capillaries surrounding organs account for 95% of the adult blood vessels and must be considered when simulating the actual characteristics of human tissue in three-dimensional (3D) microenvironments [3]. In such characteristics, it is crucial to account for factors that induce cell morphogenesis, migration, angiogenesis, metastasis and differentiation [4].

In vitro microvessel environmental systems are mainly applied by an extracellular matrix (ECM) and ECs. Pre-clinical trial models based on animals and two-dimensional (2D) in vitro models have been dominant in the field of translational research despite their limitations. Unfortunately, the differences between human taxonomy and inaccurate simulations of 3D physiologies often lead to results vastly different from actual clinical trials. However, in recent years, significant efforts have been exerted to develop artificial chip-based 3D models using microsystem technology with microfluidic devices to overcome the abovementioned flaws and to closely mimic the physical environment [4,5]. Recently, scaffolds with biological activity, such as bone morphogenic proteins (BMPs), have been utilized in tissue engineering. Additionally, various materials, including polymers, hydrogels, metals, ceramics, and bio-glass, are being explored for their potential applications in this field [6]. Microenvironmental systems mimicking functional tissue are recognized as 3D cell culture models, providing considerable advantages over well plate 2D cell culture approaches. Combining 3D cell culture in vitro assays with microenvironment-mimicking systems is effective for cell-based drug screening and the toxicity screening of chemicals to check for cell migration, angiogenesis, metastasis, and morphogenesis. The microfluidic platform has been used in many kinds of applications of 3D cell culture including 3D microvascular networks and the physiology of human lungs. The angiogenic structure of networks of ECs in hydrogel was confirmed as 3D and as regulated by the surface treatment of microfluidic channels [5]. Angiogenesis plays a crucial role in wound healing and tumorigenesis [7,8]. Tumorigenesis is mediated by TGF-β/Smads signaling, which involves vascular endothelial growth factor (VEGF), fibroblast growth factor-1 (FGF-1), platelet-derived growth factor (PDGF), and TGF-β1 [9]. TGF-β1 signaling has been shown to regulate angiogenic factors. TGF-β1 binds to TGF type I receptors and is activated by activin receptor-like kinase (ALK1) and/or activin receptor-like kinase 5 (ALK5). TGF-β1/ALK5 signaling regulates the expression of tumor promoting, suppressing, and angiogenesis genes via Smad2/3, Smad4, and FOXO1 [10,11].

Filamentous bacteriophages (*M13, F1, f88*, and *fd*) are members of the *Inoviridae* family. Filamentous bacteriophages consist of five coat proteins, approximately 2700 copies of the major coat protein pVIII, and about 3–5 copies each of the minor coat proteins pVII, pIX, pVI, and pIII per phage [12]. The f88 phage consists of two pVIII genes in one bacteriophage genome. [13]. The first major coat protein pVIII gene comprises about 2700 overlapping copies, while the second pVIII gene is randomly displayed on 10–100 copies per f88 phage [13]. Bacteriophages are being rapidly developed and utilized as functional biomaterials across various fields, including SARS-CoV-2 vaccines [14], molecularly imprinted polymers [15], lithium-ion batteries [16], SPR electrochemical biosensors [17], polyethyleneimine drug carriers [18], monoclonal antibody production (phage display libraries) [19], TLR9 signal pathway cancer therapy [20], and carbon nanofibers [21].

The advantages of bacteriophages are their ease of genetic engineering, resistance to environmental changes (temperature, pH), and biosafety for human use, as they exclusively infect bacteria. Clinical trials are currently exploring their potential in vaccines and immunotherapy [22,23,24]. The filamentous bacteriophage has a diameter of about 6.6 nm and a length of 100 nm [25], and is thereby larger than hydrogel pores which have a size of about 1–38 nm [26].

Our hypothesis is that, if the genetically modified filamentous bacteriophage encodes a growth factor and cell adhesion motif, this would not only support the ECM structure for the building block, but may also induce cell growth. In this study, we displayed TGF-β1 and the integrin-binding motif, RGD peptide, on the f88 bacteriophage. We confirmed the use of phages as an in vitro material to induce the migration and stability of ECs on a DAX-1 microfluidic chip.

## 2. Materials and Methods

### 2.1. Amplification and Purification of Phages

The f88 phage vector (kindly provided by Prof. Dr. Georg P Smith, University of Missouri, USA) was engineered to display a cyclic RGD peptide (CRGDGRC; Cys-Arg-Gly-Asp-Gly-Arg-Cys) on its pVIII major coat protein [12,13]. To insert the cyclic RGD gene sequence into the phage genome, oligonucleotide hybridization was performed whereby oligonucleotide was mixed and complementary pairs were incubated for 10 min at 100 °C. Thereafter, the chamber was slowly cooled to 50 °C over 90 min. AP-treated hybridization products were inserted into the f88 vector by *HindIII* and *KpnI* enzyme digestion followed by ligation. Active TGF-β1 (NCBI accession number: NM_000660, human TGF-β1 gene cDNA clone, Sino biological, Beijing, China) was displayed on the minor coat protein p3. TGF-β1 was cloned. The ligated vector was transferred into MC1061 *E. coli*-competent cells by electroporation (voltage: 2.5 kW; capacity: 25 µF; resistance: 200 Ω, Biorad, Feldkirchen, Germany). Subsequent to confirming the DNA sequence by sequencing analyses (Eurofin MWG operon, Ebersberg, Germany), the genetically modified phage vector was transformed into K91BK *E. coli*-competent cells to amplify the phage particles. Phage purification was performed in 20% Polyethyleneglycol (mw 8000)/2.5 M sodium chloride solution after growing the transformed K91BK cells as previously described [27]. To induce overexpression of the second recombinant pVIII genes, we added 0.5 mM of IPTG 0.5 mL to the bacterial culture medium where a cyclic RGD peptide phage was produced as in previous research [28]. To ensure minimal PEG contamination, the final soluble phage solutions were dialyzed against MilliQ water overnight at 4 °C using D-Tube Dialyzer Maxi (Millipore, Burlington, MA, USA) with a 6–8 kDa molecular weight cutoff [29].The final soluble phage supernatant was dialyzed against MilliQ water overnight to remove the remaining PEG. The concentration (colony-forming unit per milliliter, CFU/mL) of filamentous bacteriophages was determined by phage titration as previously described [15,30]. Phage suspensions were stored at 4 °C. The primer sequences used are listed in Table A1.

### 2.2. Enzyme-Linked Immunosorbent Assay (ELISA)

The concentration of TGF-β1-displaying phage was investigated with ELISA. We serially diluted with phage (10^12^–10^8^ cfu mL^−1^) or TGF-β1 (10–10^−3^ ng mL^−1^) solutions in a coating buffer (0.1 M Na_2_CO_3_, 0.1 M NaHCO_3_, pH 9.4) and coated in a 96-well ELISA plate (Thermo Scientific, Waltham, MA, USA) overnight at 4 °C. Afterward, the solution was washed three times with 200 μL of PBS containing 0.05% Tween20 (PBS-T). The plates were blocked in 2% (*w*/*v*) bovine serum albumin (BSA) (Sigma Aldrich, Baden-Württemberg, Germany) in PBS-T for 2 h at room temperature. The anti-TGF-β1 rabbit monoclonal antibody (diluted 1:1000, Abcam, Cambridge, UK) was incubated with 1% BSA in PBS-T for 1 h at 37 °C. After washing three times with PBS-T, the secondary antibody of 100 μL of peroxidase-conjugated (HRP) goat anti-rabbit IgG (diluted 1:20,000 in PBS-T containing 1% BSA, Abcam, Cambridge, UK) was added to each well and incubated for 1 h at 37 °C. After three rounds of washing, the plates were developed in ortho-phenylenediamine (Sigma Aldrich, St. Louis, MO, USA) for 5 min. The reaction was stopped by adding 50 μL of 1M H_2_SO_4_ per well. We measured the HRP substrate on a microplate reader (SPECTRA Rainbow, Tecan, Männedorf, Switzerland) at OD_492_ nm. As a negative control, we used non-genetically modified f88 bacteriophage.

### 2.3. Fluorescence Assay

To test the release of growth factor, we prepared a collagen type I gel (5 mg/mL, Ibidi, Gräfelfing, Germany). The collagen was diluted with DI water, endothelial cell growth medium (Promocell, Heidelberg, Germany), 1N NaOH, NaHCO_3_, bacteriophage (10^11^ cfu mL^−1^), and/or TGF-β1 (1 ng mL^−1^) to achieve a final concentration of 2 mg/mL with a pH of 7.4. The gel was filled (100 μL into each well) into the wells of a black 96-well microplate (Greiner Bio-One, Frickenhausen, Germany). Then, it was incubated in a humid chamber at 37 °C for 30 min. After gelation, we added 100 μL of growth-factor-free HUVEC growth media into each well. All media were changed daily. After 4 days, the collagen type I gels with TGF-β1-displaying phages and/or TGF-β1 were fixed with 4% paraformaldehyde (PFA, Sigma Aldrich) for 15 min at room temperature. Then, we blocked these with 2% BSA in PBS overnight at 4 °C. The fixed hydrogels were treated with anti-TGF-β1 rabbit monoclonal antibody (diluted 1:500) for 1 h. After washing three times with PBS, 100 μL of Alexa Fluor^®^ 488 conjugated goat anti-rabbit IgG (Abcam, Cambridge, UK) diluted 1:200 in PBS-T containing 1% BSA (Sigma Aldrich, Baden-Württemberg, Germany) was added to each well and incubated for 1 h at 37 °C. We measured fluorescence intensity on a modulus microplate multimode reader (Promega, Mannheim, Germany) at room temperature and an excitation wavelength of 490 nm and emission wavelengths of 510–570 nm, respectively.

### 2.4. Cell Culture

Human umbilical vein endothelial cells (HUVECs, Promocell, Heidelberg, Germany) were grown in endothelial cell growth medium containing 2% fetal calf serum (FCS; Promocell, Heidelberg, Germany) at 37 °C in the presence of 5% CO_2_. The cellular density and viability of HUVECs were maintained by microscopic observations and using a hemocytometer after trypan blue staining. The HUVECs were used in passages 4–6.

### 2.5. Biocompatibility Test

To identify biocompatibility, we measured cell viability by using WST-1 assays which detected the residual mitochondrial activities of HUVECs at 24 h, 48 h, 72 h, and 96 h. Briefly, 1 × 10^4^ of HUVECs were seeded in a 96-microwell plate (Thermo Scientific, Massachusetts, USA). After attaching the HUVECs, 10^11^ cfu mL^−1^ of TGF-β1-displaying bacteriophages or 10^11^ cfu mL^−1^ of f88 bacteriophages or 1 ng mL^−1^ of TGF-β1 were added to the wells of a 96-well microwell plate (Thermo Scientific, Massachusetts, USA) at 37 °C in the presence of 5% CO_2_. An amount of 10 μL of the WST-1 reagent was added to each well and monitored over 1 h on a microplate reader at OD_440_ nm. After measuring the HUVECs, we washed them three times with endothelial cell growth medium and continued the culture until the 96 h point.

### 2.6. Tube Formation Assay

To identify the cell and growth factor effects, we tested tube formation in the cells cultured on a matrigel matrix (Corning, New York, NY, USA). Briefly, 50 μL of matrigel per well was added to the FC black 96-well imaging plate (Mobitec, Goettingen, Germany). Then, the imaging plate was incubated in a humid chamber at 37 °C for 30 min. After gelation, we added 100 μL of 15,000 HUVECs with TGF-β1-displaying bacteriophages (10^11^ cfu mL^−1^) or bacteriophages (10^11^ cfu mL^−1^) or TGF-β1 (1 ng mL^−1^) to each well. The plate was incubated for 16 h. Next, the matrigel and cells were fixed with 4% paraformaldehyde for 15 min at room temperature. After fixation, we blocked them with 2% BSA in PBS overnight at 4 °C. For immunofluorescence analysis, the cell nuclei were stained with 4′,6-diamidino-2-phenylindole (DAPI; Sigma-Aldrich Baden-Württemberg, Germany) and then observed under a confocal microscope (LSM 900, Carl Zeiss, Oberkochen, Germany).

### 2.7. Lap-on-a-Chip Migration Assay

The cell migration was studied in a Lap-on-a-chip (DAX-1, 3D cell culture chip, Singapore). The hydrogel channels of a collagen type I gel were loaded either with 10^11^ cfu mL^−1^ of TGF-β1-displaying bacteriophages or the 10^11^ cfu mL^−1^ of f88 bacteriophages or 1 ng mL^−1^ of TGF-β1. The 3D cell culture chip was incubated at 37 °C in the presence of 5% CO_2_ for 30 min. After gelation, all channels were activated by adding endothelial cell growth medium for 30 min. Subsequently, one channel was seeded with 2 × 10^6^ HUVECs. The media were changed every 12 h. For immunostaining, we used the three dyes DAPI, CD 31 mouse monoclonal antibody (1:1600, Cell Signaling Technology, Danvers, MA, USA), and VE-Cadherin (1:400, Cell signaling Technology) and observed them under a confocal microscope.

### 2.8. RT-qPCR

To identify gene expression patterns, HUVECs were cultured with 10^11^ cfu mL^−1^ of TGF-β1-displaying bacteriophages or 10^11^ cfu mL^−1^ of f88 bacteriophages or 1 ng mL^−1^ of TGF-β1 for 4 days. The phage-free media were changed every 12 h. After 4 days, the cells were harvested by using trypsin-EDTA (0.25%, Thermo Fisher scientific, Frederick, MD, USA). Total RNA extraction from the harvested cells was performed by using Trizol (Gibco BRL, Grand Island, NY, USA) according to the manufacturer’s instructions. The concentration of total RNA was measured by the Nano drop (Thermo Fisher scientific). cDNA synthesis was performed by a high-capacity RNA to cDNA kit (applied biosystems, Thermoscientific). Primer- and fluorescent dye-labeled TaqMan MGB probes were designed based on gene bank sequences for CD31 (accession number M37780), VE-cadherin (accession number U84722), and GAPDH (accession number NM_002046) by using the PRIMER EXPRESS 3.0 program (Applied Biosystems, Foster City, CA, USA). The real-time PCR was carried out with one cycle of pre-denaturation at 94 °C for 5 min, followed by 40 cycles of amplification with denaturation for 30 s at 94 °C, annealing for 30 s at 60 °C, extension for 30 s at 72 °C, and a final 10 min extension at 72 °C in a 7500 fast real-time PCR system (Applied Biosystems, Foster City, CA, USA).

### 2.9. Statistical Analysis

All the data were analyzed using the software SigmaPlot 12.0 (SigmaPlot Software, La Jolla, CA, USA). The quantitative data are expressed as mean ± standard error of the mean (SEM). From Figure 1, Figure 2, Figure 3, Figure 4 and Figure 5, statistical significance was analyzed by using the Kruskal–Wallis and Mann–Whitney tests, where differences were considered statistically significant at a *p*-value < 0.05. While all experiments were performed in triplicate, only the data presented in Figure 6 yielded statistically significant results. Statistical analyses were conducted for the other figures and tables; however, the results did not pass the significance threshold (*p*-value ≥ 0.05). Consequently, these data are presented as trends, reflecting the experimental outcomes as observed without statistical significance.

## 3. Results

As shown in Figure 1, we confirmed the induction of an invasion of ECs due to genetically engineered bacteriophages using a DAX-1 microfluidics chip. We constructed filamentous bacteriophages based on f88 phages that express recombinant human TGF-β1 cloned into the major coat protein pIII and the cell adhesion motif (cysteine mediate RGD peptide) cloned into the major coat protein pVIII (Figure 1a). The f88 phages sparsely displayed the major coat protein which carried 10 to 100 copies of recombinant pVIII proteins. The filamentous bacteriophage vectors were provided by the Smith group (Prof. Dr. Georg P Smith, University of Missouri, USA). We used the DAX-1 microfluidic chips which commercialized 3D multicellular cell culture. DAX-1 comprises three multichannels which provide one polymerizable gel and two of medium channels (Figure 1b). As shown in Figure 1c–e, we seeded the hydrogel, bacteriophages, and HUVECs in the channels. After seeding, we monitored the HUVEC invasion of microfluidic chips (Figure 1f).

**Figure 1 jfb-15-00314-f001:**
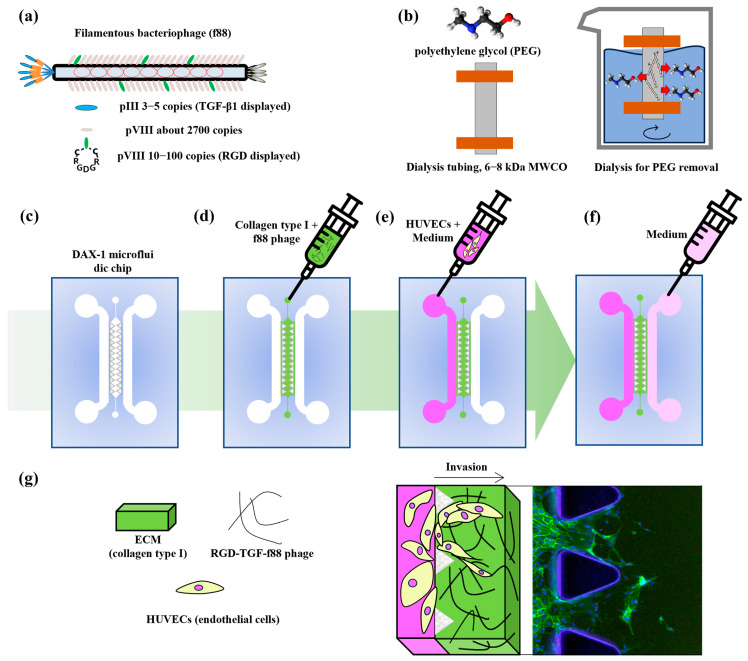
Schematic illustration of the construction of the functional filamentous bacteriophages and cell seeding and cell culturing in microfluidic chips. (**a**) The structure of the functionalized genetically engineered site of the filamentous bacteriophages. (**b**) Dialysis for removal of contaminants and unbound PEG from the bacteriophage elution solution. (**c**) The microfluidics chip of DAX-1. (**d**) The polymerization of the type I hydrogel and bacteriophages at the gel channel. (**e**) The seeding of cells with medium at the left medium channel. (**f**) The filling of the medium at the right medium channel. (**g**) The monitoring of the HUVEC invasion during TGF-b1 interaction with cells in a microfluidics chip.

### 3.1. ELISA Results

To test the relative amount of the TGF-β1 recombinant at the pIII minor coat proteins, we performed the enzyme-linked immunosorbent assay (ELISA). In Figure 2, absorbance intensity represents the TGF-β1 proteins that validate the quantification of the protein on 96-well plate surfaces. We calculated the relative amount of TGF-β1 between the displayed phages and proteins by setting the wild-type f88 filamentous bacteriophage as the control. In order to optimize the relative concentration of RGD-TGF-β1 bacteriophages, we plotted the logistic regression model in Table 1. As shown in Figure 2 and Table 1, the fitted ELISA curves had high regression coefficients (r2) which ranged from 0.9974 to 0.999, and their ANOVA *p*-values were smaller than 0.0026.
Logistic L: Ai absorbance intensity=α/1+c/γ^β

As shown in Figure 2, regression curves between TGF-β1 and the RGD-TGF-β1 phage had one crossing point at a 0.25 absorbance intensity. TGF-β1 exhibited an absorbance intensity of 0.2482 ± 0.0347 at a concentration of 1 ng mL^−1^. The RGD-TGF-β1 phage had a concentration of 0.2407 ± 0.0306 at a 1011 pfu/mL concentration.

**Figure 2 jfb-15-00314-f002:**
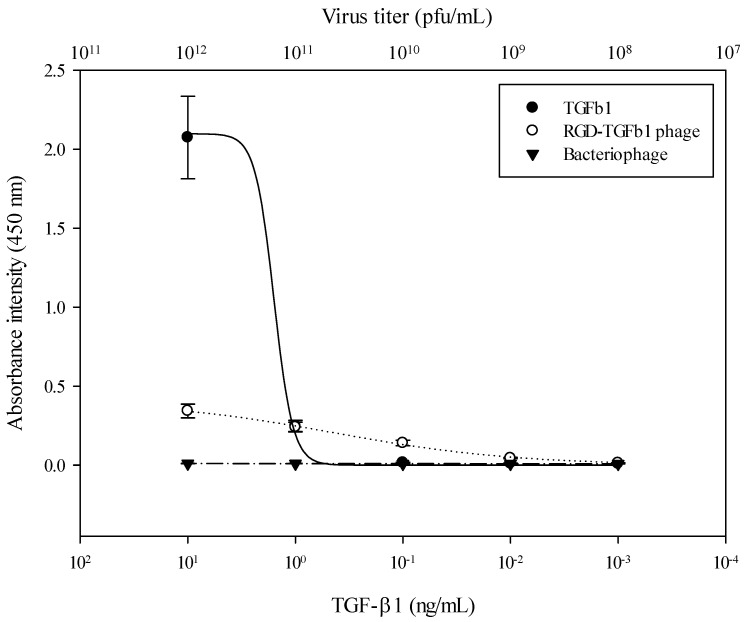
The evaluation of the TGF-β1 proteins on genetically modified bacteriophages by ELISA. Data are presented as mean ± SEM.

### 3.2. Fluorescence Results

As shown in Figure 3, a collagen type I gel with embedded growth factors was confirmed by immunostaining the fluorescence-labeled TGF-β1 proteins. We hypothesized that macromolecules of the RGD-TGF-β1 are displayed on the bacteriophages. The TGF-β1-sensitive fluorescence intensities were measured on day 0 as being 2.9 × 10^3^ ± 3.2 × 10^1^ of 1 ng of TGF-β1 and 3.4 × 10^3^ ± 2.8 × 10^2^ of 1011 pfu mL^−1^ of RGD-TGF-β1 phages, and on day 4 as being 2.5 × 10^3^ ± 3.1 × 10^2^ of TGF-β1 and 3.4 × 10^3^ ± 1.5 × 10^2^ of RGD-TGF-β1 phages. The results show that the macromolecules of the filamentous bacteriophages remained in a collagen matrix. However, the native TGF-β1 proteins were transferred into the medium. In general, the cell culture medium was replaced every 2–3 days. Some of the cells require a growth factor when cultivated. In particular, if we create a 3D microenvironment with hydrogels, we validate that the growth factors have integrated into the hydrogels and track the mode of action to the effects on the cells. However, suppose the RGD-TGF-β1 filamentous bacteriophages existed in the collagen matrix. In that case, the displayed growth factor can affect cellular growth for a long-term cell culture because the RGD-TGF-β1 bacteriophages are larger in size than the hydrogel pores [25,26]. This observation suggests that bacteriophages can probably be used in long-term cell culture systems.

**Figure 3 jfb-15-00314-f003:**
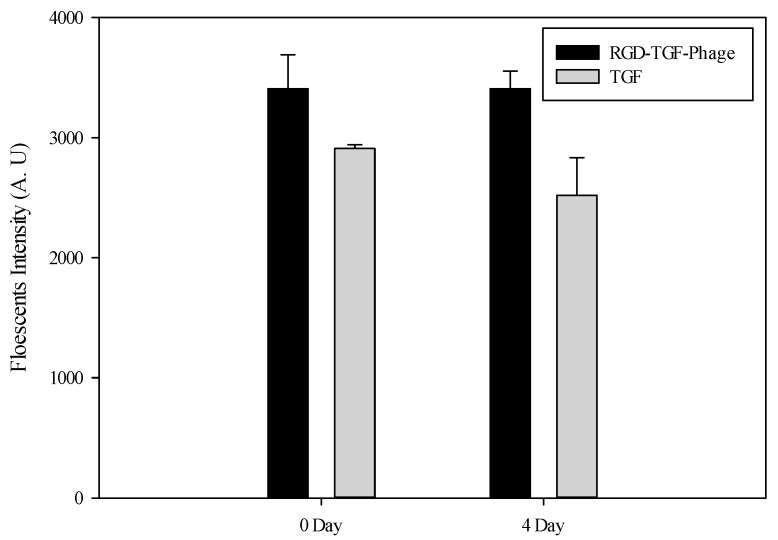
The fluorescence intensity profiles of phages and growth factor in the type I collagen (Days 0 and 4). Data are presented as mean ± SEM.

### 3.3. Viability Test

Next, we characterized the cell viability against genetically modified filamentous bacteriophages. HUVECs were seeded onto a 96-well plate at concentrations of 2 × 10^4^ cells·cm^−3^. After 24 h, we assessed the presence of growth factors and/or bacteriophages on a culturing plate. After 72 h of culture, we evaluated mitochondrial activities by WST-1 assay. For comparison of cell growth properties, we selected FBS including growth medium as the positive control and FBS without additional growth reagents as a negative control. As shown in Figure 4, the resulting viabilities are 140% for the TGF-β1-treated positive controls, 169% for the TGF-β1-displaying bacteriophages, 103% for the natural bacteriophages, and 4% for the negative control after 72 h of culture. This experiment confirmed that ECs grew faster when TGF-β1 was added than without TGF-β1 in the medium. Additionally, this shows that filamentous bacteriophages did not exhibit an adverse effect on the cell viabilities. Therefore, this observation suggests that endothelial cell cultures could be co-cultivated with filamentous bacteriophages over 4 days of culture.

**Figure 4 jfb-15-00314-f004:**
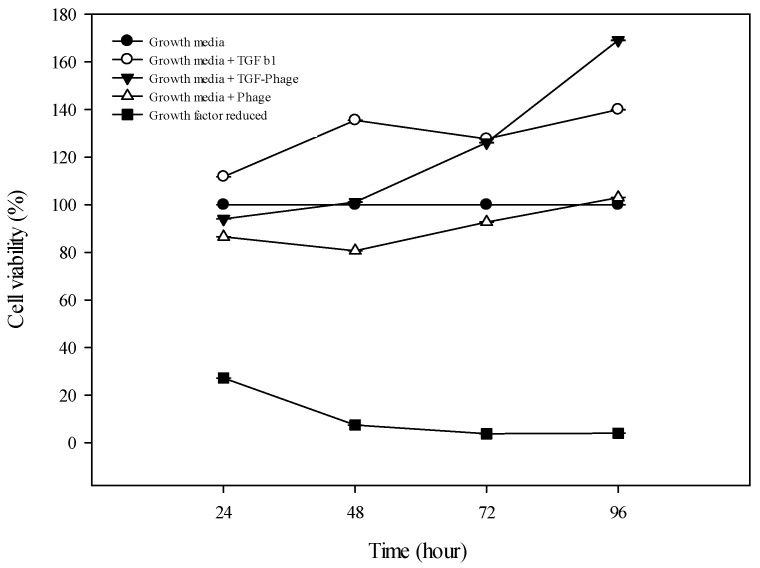
The proliferation of HUVECs, cultured with bacteriophages (Days 0, 1, 3, and 4), was determined by a CCK-8 cell proliferation assay. Data are presented as mean ± SEM.

### 3.4. Tube Formation of ECs

To define the network pattern of competitive ECs, we performed a tube formation assay on matrigel. Figure 5 shows that, according to growth factor and/or bacteriophage treatment conditions, either pseudo-capillary growth and/or branched points were observed. The pseudo-capillary growth and branched points in Figure 5e,f were evaluated by using the software ImageJ version 1.54 [7]. The RGD-TGF-β1 bacteriophages generated 5.40 ± 0.29 mm of total capillary extension and 14.7 ± 0.58 branch points (Figure 5d). In comparison, the positive control of TGF-β1 growth factor generated 5.14 ± 0.31 mm of total capillary extension and 13.67 ± 1.53 branch points (Figure 5b), whereas the negative control of the growth factor had a reduced total length of 4.05 ± 0.54 mm with 9.0 ± 1.73 branch points. Hence, this analysis showed that TGF-β1 growth factor induced an enhanced extension of pseudo-capillary ECs. Interestingly, non-displaying bacteriophages were weakly stimulated and induced tube formation with a total length of 4.96 ± 0.33 mm, and 11.33 ± 1.53 branch points in comparison with the negative control of Figure 5a. As shown in Figure 4, the RGD-TGF-β1 bacteriophage was found to be safe and growth-supporting. Furthermore, Figure 5 shows the efficacy of the bacteriophage in tube formation. In summary, these data confirm that RGD-TGF-β1 bacteriophages can be used as a biomaterial to induce a microvessel formation model using ECs during the construction of a microenvironment.

**Figure 5 jfb-15-00314-f005:**
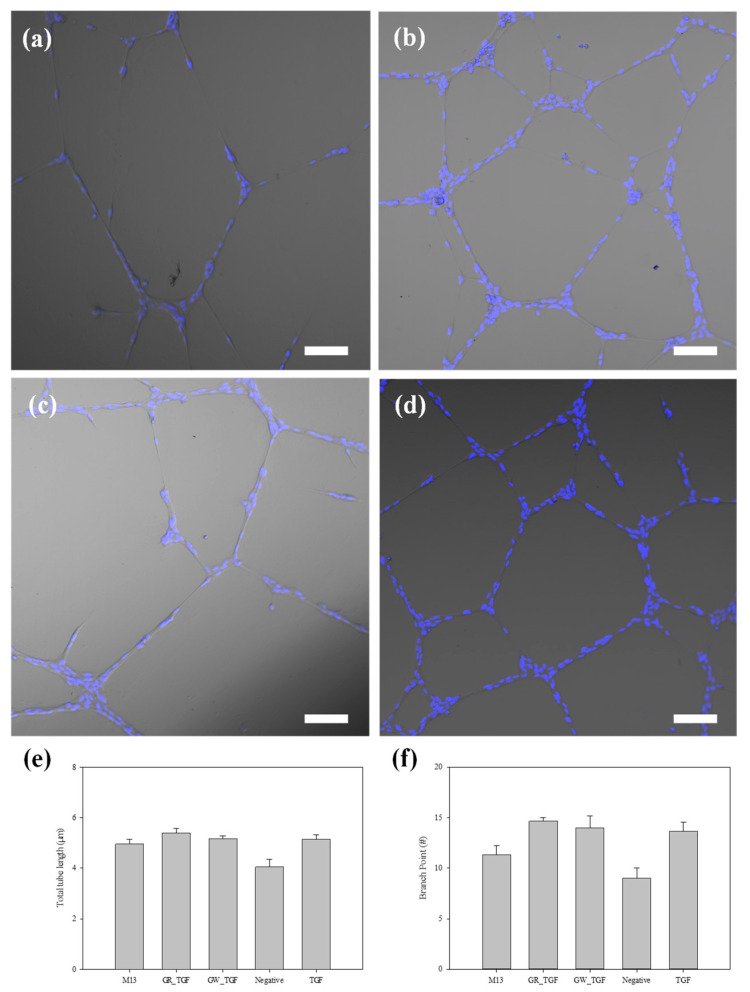
Tube formation of HUVECs on matrigel after 24 h of seeding; (**a**) negative control; (**b**) TGF-β1, 1 ng mL^−1^; (**c**) f88, 1 × 10^11^ cfu mL^−1^; (**d**) RGD-TGF phage, 1 × 10^11^ cfu mL^−1^. The image analysis was carried out with the angiogenesis analyzer module of ImageJ; (**e**,**f**) show the total tube length (**e**) and the number of branch points (**f**). The scale bar in (**a**–**d**) is 100 μm (40× magnification). Data are presented as mean ± SEM.

### 3.5. Migration Test on a Microfluidics Chip

We used the DAX-1 microfluidics chip designed by AIM biotech to generate a migration microenvironment. It has three channels in total, two channels for cell and/or media and one central gel channel. We generated the hydrogel of collagen type I in a gel channel and seeded the ECs in a media channel. As shown in Figure 6, the ECs showed migration effects into the hydrogel. The average migration length was determined as 200.84 ± 76.29 μm for the group treated with RGD-TGF bacteriophages and 149.98 ± 75.82 μm for the group directly induced by TGF-β1 growth factor. In comparison, the native filamentous bacteriophages used as a negative control showed only a 74.61 μm migration length. We stained two endothelial cell biomarkers and the nucleus with CD31 (green), VE-cadherin (red), and DAPI (blue; nucleus). The results showed a strong CD31 expression. However, the cell-to-cell adhesion junction of VE-cadherin (CD144) was not detected in ECs in Figure 6a–d. To quantify expression, we tested the mRNA levels of CD31, VE-cadherin, and the housekeeping gene GAPDH by TaqMan qRT-PCR. As shown in Figure 6f, the CD31 mRNA levels increased five-fold when TGF-β1 growth factor was induced. However, VE-cadherin expression was not detected at the mRNA level when induced by TGF-β1 growth factor. These results show that the mRNA level of CD31 from qPCR was correlated with the results from the confocal microscope. In general, while HUVECs upregulated VE-cadherin, Ando et al. and Kocherova et al. showed that shearing stress and/or medium condition can reduce the level of VE-cadherin [31,32]. Interestingly, the tube formation and the number of branch points tended to increase compared to the negative control in the natural f88 page used as the control of the bacteriophage, and similar results were obtained in the fluidics chip as shown in Figure 5c and Figure 6c. This proves that the linear structural characteristics of bacteriophage affected endothelial cell migration.

**Figure 6 jfb-15-00314-f006:**
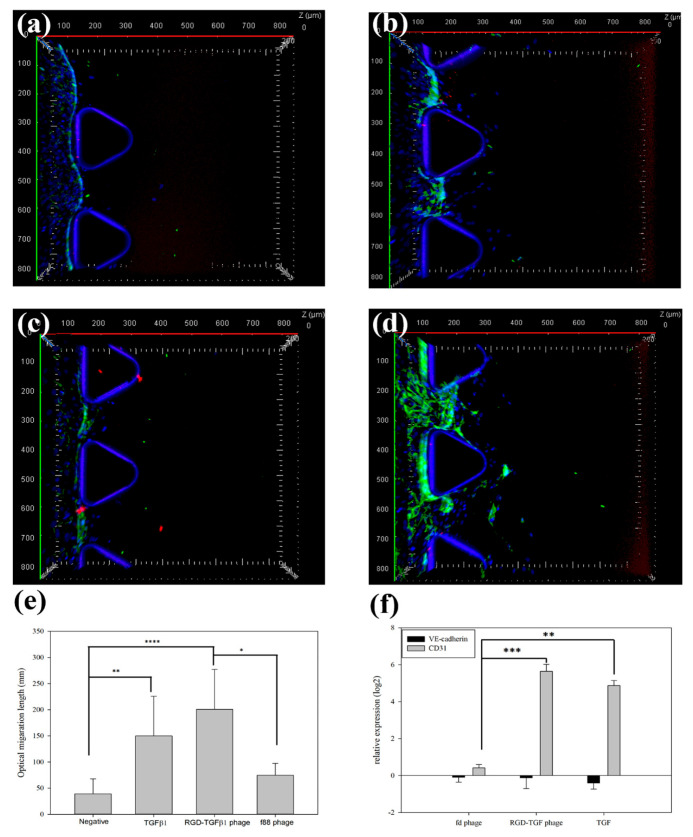
The observation of angiogenesis on a vascular-on-a-chip after 4 days of treatment. The conditions are as follows: (**a**) growth media control, (**b**) TGF-β1 (1 ng/mL), (**c**) f88 phage (1 × 10¹¹ cfu/mL), and (**d**) RGD-TGF phage (1 × 10¹¹ cfu/mL). (**e**) The quantification of the optical migration length (mm) and (**f**) mRNA expression levels of CD31 and VE-cadherin, measured by real-time PCR. Data are presented as mean ± SEM. Statistical analysis was performed using the Kruskal–Wallis test with a confidence interval of *p* < 0.04 (*n* = 6). Significant differences are indicated by *p*-values of <0.05 (*), <0.01 (**), <0.001 (***), <0.0001 (****), respectively.

## 4. Discussion

According to the 3R principles (replacement, reduction, and improvement), it is generally desirable to replace animal testing by alternative non-animal testing methods. As alternative testing methods, 3D cell culture, organoid, and lab-on-a-chip methods are being developed. In this study, we verified the function and effectiveness of genetically modified filamentous bacteriophages as safe in vitro models in tissue engineering. We showed that the genetically manipulated phages can sustainably support the growth of cells in a long-term culture. Three-dimensional cell culture is a technology used to mimic organ and tissue culture. During long-term 3D cell culture, there is often a lack of proper circulation of nutrients and oxygen, so that the inner cells may turn necrotic. Thus, we utilized a lab-on-a-chip platform that facilitates convenient media exchange, allowing for better control of oxygen levels [33]. To overcome this problem, previous studies modifying the ECM, cells, and media have been reported [34,35]. Furthermore, our results suggest that TGF-β1 displayed by bacteriophages functions as a chemotactic factor, promoting endothelial cell migration and angiogenesis. Although we did not directly observe a physical interaction between the ECs and the bacteriophages, the increase in cell migration and tube formation observed is consistent with the known roles of TGF-β1 in stimulating chemotactic responses, as evidenced by CD31 expression in ECs. This effect may further enhance cell adhesion and migration in the presence of RGD. We believe that our study is closely related to concepts modifying the ECM, and provides additional functionality as a support for cell culture. This study confirmed the safety and long-term effectiveness of the potential materials. We did not observe abnormalities in the cells. Bacteriophage is a material that proves biosafety and is currently undergoing phase 2 clinical trials [36]. Using bacteriophages emphasizes safety when used as materials; although, the simple, functional increases are also substantial. Additionally, we employed bacteriophages ranging from 6.6 nm to 100 nm in size, which are larger than the hydrogel pore sizes (1–38 nm), to enhance interaction with the ECM. Our study demonstrated that TGF-β1 displayed by the bacteriophages promoted angiogenesis and endothelial cell (EC) tube formation, validating their role in enhancing vascular formation. However, further research is needed to fully understand the potential of bacteriophages as ECM biomaterials and their long-term effects on angiogenesis. Moreover, our results suggest that TGF-β1 displayed by bacteriophages serves as a chemotactic factor, enhancing endothelial cell migration and promoting angiogenesis. Although we did not directly observe a physical interaction between ECs and the bacteriophages, the increase in cell migration and tube formation aligns with TGF-β1’s role in stimulating chemotactic responses, as evidenced by CD31 expression in ECs as previously described [37]. This effect could potentially enhance cell adhesion and migration, particularly in the presence of RGD. One of the greatest challenges in tissue engineering is achieving long-term cell culture viability by mimicking the oxygen and nutrient delivery systems found in vivo [33]. By utilizing genetically modified filamentous bacteriophages, we demonstrate a promising alternative that not only supports cell migration but also supports growth factor in long-term cultures of 3D cell culture systems. The growth factor-displaying bacteriophages enhance angiogenesis and endothelial cell migration, key components of vascularization. Compared to traditional cell culture, bacteriophages offer a level of biosafety and adjustability that makes them ideal for use in scaffolds [38]. Despite these findings, the risk of endotoxin contamination must be carefully addressed in future studies, as highlighted by previous research [39,40]. Effective control of endotoxin mechanisms will be crucial to ensure the safety and reproducibility of bacteriophage-based biomedical applications.

One key step in cancer metastasis is that tumor cells penetrate into tissues and induce angiogenesis. Both stages are regulated by the interaction between the tumor microenvironment and cell-to-cell interaction [41,42]. When trying to mimic this microenvironment in an in vitro system, it is important to supply nutrients and growth factors. We would like to reduce the gap to in vivo cancer metastasis conditions by using a microfluidic platform to mimic the tumor microenvironment and cell-to-cell interaction in vitro. In fact, microfluidics platforms have been developed previously to mimic angiogenesis and organ-on-chip implementation [43,44,45]. Three essential components are required for the implementation of 3D cell culture. One needs the cells to be implemented, the supports to support them, and the necessary nutrients so that the cells grow and function properly. Many studies are ongoing and attempts are being made to create an organ-like environment. In line with these studies, we propose a new material that can provide a continuous inflow of nutrients using genetically modified filamentous bacteriophages that display growth factors and increase the permeability of ECM to help with long-term incubation. We advocate the possibility of using bacteriophage displays as a safe material to independently monitor biochemistry and components to study cell-to-cell interactions. In our previous work, we reported on the use of VEGF-A bacteriophages in microfluidic chips [38]. With the continuous supply of VEGF-A and TGF-b1 growth factors and induction of vascular production, an organ-on-a-chip will be constructed by building a cancer metastasis model on the microfluidics chip through co-culture with the organ-specific cells. Additionally, Doub et al. have reported that the use of bacteriophages can lead to endotoxin contamination and the formation of bacterial biofilms [39,40]. Despite these findings, the risk of endotoxin contamination must be carefully addressed in future studies, as highlighted by previous research [39,40]. Effective control of endotoxin mechanisms will be crucial to ensure the safety and reproducibility of bacteriophage-based biomedical applications. Thus, to better understand the efficacy of bacteriophages, it is crucial to implement controls for endotoxin and bacterial enzyme contamination in future studies. In the future, we plan to implement a platform that can assess the effectiveness of in vitro drugs, which has only been found in animals through our research.

## 5. Conclusions

This study investigated the potential use of an angiogenic matrix in tissue engineering as an alternative to conventional in vitro models. The researchers observed that ECs established connections with biocompatible bacteriophages, which were able to migrate and sprout into the ECM. The findings indicated that bacteriophages displaying biocompatible TGF-β1 could contribute to the continuous stimulation of the microenvironment, promoting angiogenesis in in vitro models. Furthermore, the researchers suggested that these functionalized bacteriophages could serve as a viable biomaterial in biomedical engineering. The study demonstrated that growth factor-displaying filamentous bacteriophages could be utilized not only for long-term cell culture, providing biocompatible and continuously effective materials, but also in other biomedical engineering applications such as 3D cell culture and organoid culture. Our results propose that similar strategies could be employed in the future to enhance angiogenic matrixes in tissue engineering assays.

## Figures and Tables

**Table 1 jfb-15-00314-t001:** The parameters of the logistic models for TGF-β1, RGD-TGF-β1 phage, and bacteriophage.

Materials	*r* ^2^	Logistic Model Parameters		
		Height	Slope	Slop at Max 50%
TGF β1	0.9987	2.0984	−4.9648	1.5817
RGD-TGF-β1 phage	0.9974	0.4074	−0.5181	0.4317
Bacteriophage	0.9999	0.0166	−0.0395	0.0012

## Data Availability

The original contributions presented in the study are included in the article, further inquiries can be directed to the corresponding author.

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
