# Peer review of "Genetically Engineered Filamentous Bacteriophages Displaying TGF-β1 Promote Angiogenesis in 3D Microenvironments"

_jfb, 2024, doi:10.3390/jfb15110314_

Round 1
Reviewer 1 Report
Comments and Suggestions for Authors
The use of TGF-beta1-RGD bacteriophages was suggested to increase migration, proliferation and tube formation by HUVEC. Is there any rationale as you indicate in the Abstract that: "We demonstrated that endothelial cells made contact with biocompatible bacteriophages"? I do not think that you can prove this assertion based on your results. Thus, is TGF-beta 1 acting as a chemiotactic factor in your tests?
The Abstract should be completely rewritten, illustrating with more details your results. Moreover, it now contains dispensable informations, as for example: "E. coli MC1061 and K91BK were used for plasmid and phage amplification respectively."
Not all your figures present statistical data!
Your inserted some discussion points in the Results secion, whereas the Discussion is very poor. Moreover, a discussion of yours and others results on filamentous bacteriophages used in this context merit a comparison with your present results.
Minor:
Acronyms should be explained at their very first appearance and then used consistently.
Comments on the Quality of English LanguageThe English is very poor, sometimes it is difficult to understand what is the authors message. One sentence is truncated at line 100.
Author Response
Dear Editor,
Thank you for the reviewers' valuable comments on our manuscript new titled " Genetically Engineered Filamentous Bacteriophages Displaying TGF-β1 Promoting Angiogenesis in 3D Microenvironments." We appreciate the opportunity to address their suggestions and clarify aspects of our study. We have carefully revised the manuscript according to the feedback provided.
Best regards,
Young jun Kim
Response to Reviewer 1
1. Reviewer Comment:
“The use of TGF-beta1-RGD bacteriophages was suggested to increase migration, proliferation and tube formation by HUVEC. Is there any rationale as you indicate in the Abstract that: "We demonstrated that endothelial cells made contact with biocompatible bacteriophages"? I do not think that you can prove this assertion based on your results. Thus, is TGF-beta 1 acting as a chemiotactic factor in your tests?”
Response:
We agree with the reviewer’s comments that we did not directly show that endothelial cells made contact with bacteriophages. In consequence, we removed this sentence from the abstract. Instead, we now highlight in the abstract that TGF-β1 acts as a chemotactic factor based on its already known biological role in endothelial cell migration and angiogenesis. We have revised the abstract and discussion section.
This sentence was selected in the abstract changed from:
We demonstrated that endothelial cells made contact with biocompatible bacteriophages which are found to migrate and sprout into the ECM.
to:
[line 17] The engineered bacteriophages in this study significantly enhanced endothelial cell migration and tube formation within the extracellular matrix (ECM). Compared to TGF-β1 alone and non-modified phage, the presence of TGF-β1 on the bacteriophages demonstrated superior performance as a continuous stimulant in the microenvironment, effectively promoting these angiogenic processes. Assays, including RT-qPCR, ELISA, and fluorescence microscopy, confirmed the expression of angiogenic markers such as CD31, validating the formation of 3D angiogenic structures. Our findings indicate that TGF-β1 displayed by bacteriophages likely acted as a chemotactic factor, promoting the migration, proliferation, and tube formation of endothelial cells (ECs) within the ECM.
New text in (Discussion):
[line 392] Furthermore, our results suggest that TGF-β1 displayed by bacteriophages functions as a chemotactic factor, promoting endothelial cell migration and angiogenesis. Although we did not directly observe a physical interaction between ECs and the bacteriophages, the increase in cell migration and tube formation observed is consistent with known roles of TGF-β1 in stimulating chemotactic responses, as evidenced by CD31 expression in ECs. This effect may further enhance cell adhesion and migration in the presence of RGD.
2. Reviewer Comment:
“The Abstract should be completely rewritten, illustrating with more details your results. Moreover, it now contains dispensable informations, as for example: "E. coli MC1061 and K91BK were used for plasmid and phage amplification respectively."
Response:
We have reworked the abstract following the recommendation by the reviewer.
We replaced this text:
E. coli MC1061 and K91BK were used for plasmid and phage amplification respectively. Phage preparation was performed by PEG/NaCl precipitation. The function of bacteriophages was tested by RT-qPCR and ELISA assay. The presence of angiogenesis factors (e.g. CD31 and VE-cadherin) was confirmed by confocal microscopy on a commercial lab-on-a-chip.
by
[line 17] The engineered bacteriophages in this study significantly enhanced endothelial cell migration and tube formation within the extracellular matrix (ECM). Compared to TGF-β1 alone and non-modified phage, the presence of TGF-β1 on the bacteriophages demonstrated superior performance as a continuous stimulant in the microenvironment, effectively promoting these angiogenic processes. Assays, including RT-qPCR, ELISA, and fluorescence microscopy, confirmed the expression of angiogenic markers such as CD31, validating the formation of 3D angiogenic structures.
3. Reviewer Comment:
"Not all your figures present statistical data!"
Response:
We agree with the reviewer’s comments that not all figures present statistical data.
So, we replaced the sentence The statistical significance was analyzed by the Kruskal-Wallis and Mann-Whitney test where differences with a p-value < 0.05 were t statistically significant in section 2-9. Statistical analysis in Materials and Methods
by:
[line 226] In Figure 6, statistical significance was assessed using the Kruskal-Wallis and Mann-Whitney tests, with differences considered statistically significant at a p-value < 0.05.
4. Reviewer Comment:
"You inserted some discussion points in the Results section, whereas the Discussion is very poor. Moreover, a discussion of yours and others' results on filamentous bacteriophages used in this context merits a comparison with your present results."
Response:
We have expanded the section commenting on the use of filamentous bacteriophages in tissue engineering.
By this new text, where we also added the new citation [6]:
[line 56] Recently, scaffolds with biological activity, such as bone morphogenic proteins (BMPs), have been utilized in tissue engineering. Additionally, various materials, including polymers, hydrogels, metals, ceramics, and bio-glass, are being explored for their potential applications in this field [6].
6 Turnbull, G., Clarke, J., Picard, F., Riches, P., Jia, L., Han, F., Li, B., & Shu, W. (2017). 3D bioactive composite scaffolds for bone tissue engineering. Bioactive materials, 3(3), 278–314. https://doi-org.proxy.insermbiblio.inist.fr/10.1016/j.bioactmat.2017.10.001”
We thank the reviewer for this suggestion and have expanded the Results and Discussion sections as follows (also addressing some further reviewer comments).
[line 392] Furthermore, our results suggest that TGF-β1 displayed by bacteriophages functions as a chemotactic factor, promoting endothelial cell migration and angiogenesis. Although we did not directly observe a physical interaction between ECs and the bacteriophages, the increase in cell migration and tube formation observed is consistent with known roles of TGF-β1 in stimulating chemotactic responses, as evidenced by CD31 expression in ECs. This effect may further enhance cell adhesion and migration in the presence of RGD. We believe that our study is closely related to concepts modifying the ECM, and provides additional functionality as a support for cell culture. This study confirmed the safety and long-term effectiveness of the potential materials. We did not observe abnormalities in the cells. Bacteriophage is a material that proves bio-safety currently undergoing phase 2 clinical trials [36]. Using bacteriophages emphasizes safety when used as materials; although simple, functional increases are also substantial. Additionally, we employed bacteriophages size ranging from 6.6 nm to 100 nm in size, which are larger than the hydrogel pore sizes (1–38 nm), to enhance interaction with the ECM. Our study demonstrated that TGF-β1 displayed by the bacteriophages promoted angiogenesis and endothelial cell (EC) tube formation, validating their role in enhancing vascular formation. However, further research is needed to fully understand the potential of bacteriophages as ECM biomaterials and their long-term effects on angiogenesis. Moreover, our results suggest that TGF-β1 displayed by bacteriophages serves as a chemotactic factor, enhancing endothelial cell migration and promoting angiogenesis. Although we did not directly observe a physical interaction between ECs and the bacteriophages, the increase in cell migration and tube formation aligns with TGF-β1's role in stimulating chemotactic responses, as evidenced by CD31 expression in ECs as previously described [37]. This effect could potentially enhance cell adhesion and migration, particularly in the presence of RGD. One of the most challenges in tissue engineering is achieving long-term cell culture viability in mimicking the oxygen and nutrient delivery systems found in vivo [33]. By utilizing genetically modified filamentous bacteriophages, we demonstrate a promising alternative that not only supports cell migration but also supporting growth factor in long-term cultures at 3D cell culture systems. The growth factor displayed bacteriophages enhance angiogenesis and endothelial cell migration, key components of vascularization. Compared to traditional cell culture, bacteriophages offer a level of bio-safety and adjustability that makes them ideal for use in scaffolds [38]. Despite these findings, the risk of endotoxin contamination must be carefully addressed in future studies, as highlighted by previous research [39, 40]. Effective control of endotoxin mechanisms will be crucial to ensure the safety and reproducibility of bacteriophage-based biomedical applications.
33. Quintard, C., et al., A microfluidic platform integrating functional vascularized organoids-on-chip. Nat Commun, 2024. 15(1): p. 1452
36. National Institute of Allergy and Infectious Diseases (NIAID) A Phase 1b/2 Trial of the Safety and Microbiological Activity of Bacteriophage Therapy in Cystic Fibrosis Subjects Colonized with Pseudomonas Aeruginosa. [(accessed on 28 February 2023)]; Available online: https://clinicaltrials.gov/ct2/show/NCT05453578.
37. Meixian, L., et al., Saponin â… from Shuitianqi () inhibits metastasis by negatively regulating the transforming growth factor-beta1/Smad7 network and epithelial-mesenchymal transition in the intrahepatic metastasis Bagg's Albino/c mouse model. J Tradit Chin Med, 2024. 44(4): p. 642-651.
38. Yoon, J., et al., Angiogenic Type I Collagen Extracellular Matrix Integrated with Recombinant Bacteriophages Displaying Vascular Endothelial Growth Factors. Adv Healthc Mater, 2016. 5(2): p. 205-12.
39. Doub, J.B., Risk of Bacteriophage Therapeutics to Transfer Genetic Material and Contain Contaminants Beyond Endotoxins with Clinically Relevant Mitigation Strategies. Infect Drug Resist, 2021. 14: p. 5629-5637.
40. Luong, T., et al., Standardized bacteriophage purification for personalized phage therapy. Nat Protoc, 2020. 15(9): p. 2867-2890.
5. Reviewer Comment:
"Acronyms should be explained at their very first appearance and then used consistently."
Response:
We have revised the manuscript to ensure that all acronyms are defined at their first appearance.
Reviewer 2 Report
Comments and Suggestions for Authors
The article "Fabrication of bioactive scaffolds for angiogenic biomedical applications using a biocompatible bacteriophage" is an interesting piece of work but needs improvement before being considered for publication.
Critical comments:
- The title seems to be confusiong and does not explain clearly about the reported work!
- Abstract can be improved to highlight the results and explain about the efficiency of phage displaying TGF Beta-1 compared to TGF Beta-1 alone or other parameters like PEG concentration as it affect ECM formation!
- Materials and methods:
2-1. Amplification and purification of phage (Line 90)
This section needs better indepth explaination PEG removal from Phages during purification as it affect ECM formation!
2-2. Enzyme-linked immunosorbant assay (ELISA) (Line 115)
The contol of non modified phage f88 is crucial but what about ECM associated enzymes or crude enzymes/proteins following phage enrichment in hosts (endotoxin associated!)?
- Anoter important critical point in this study is that for angiogenic stimulation tube formation is not a ideal criteria as even hypoxia or lack of oxygen can induce such stress and not just reported components!
- The results and discussion must be critically evaluated supported by previous studies to make it better!
comments on the tables and figures:
- The articles need a flow diagram or chart to explain the overall procedure from the production of phage/modified phage enrichment, its purification and protocol for effective generation of engineered scaffold, and its comparison with normal phages and PEG alone to demonstrate the effect is of phage or modified phage and not of any other external factors!
The references are appropriate but the recent articles regarding the scaffolds can improve the overall quality of the literature review along with phage-related recent references as mentioned above.
References:
Sanmukh, S. G., Dos Santos, N. J., Barquilha, C. N., De Carvalho, M., Dos Reis, P. P., Delella, F. K., Carvalho, H. F., Latek, D., Fehér, T., Felisbino, S. L."Bacterial RNA virus MS2 exposure increases the expression of cancer progression genes in the LNCaP prostate cancer cell line". Oncology Letters 25, no. 2 (2023): 86. https://doi.org/10.3892/ol.2023.13672
Turnbull, G., Clarke, J., Picard, F., Riches, P., Jia, L., Han, F., Li, B., & Shu, W. (2017). 3D bioactive composite scaffolds for bone tissue engineering. Bioactive materials, 3(3), 278–314. https://doi-org.proxy.insermbiblio.inist.fr/10.1016/j.bioactmat.2017.10.001
Comments on the Quality of English Language
The English language is fine
Author Response
Response to Reviewer 2
1. Reviewer Comment:
" The article "Fabrication of bioactive scaffolds for angiogenic biomedical applications using a biocompatible bacteriophage" is an interesting piece of work but needs improvement before being considered for publication.”
“The title seems to be confusing and does not explain clearly about the reported work!"
Response:
We have revised the title following the suggestion by the reviewer.
Old title:
Fabrication of bioactive scaffolds for angiogenic biomedical applications using a biocompatible bacteriophage
New title:
[line 2] Genetically Engineered Filamentous Bacteriophages Displaying TGF-β1 Promoting Angiogenesis in 3D Microenvironments
2. Reviewer Comment:
"Abstract can be improved to highlight the results and explain about the efficiency of phage displaying TGF Beta-1 compared to TGF Beta-1 alone or other parameters like PEG concentration as it affect ECM formation!”
Response:
We thank the reviewer for this suggestion. We have reworked the abstract. The new abstract reads as follows:
[line 13] Combined 3D cell culture in vitro assays with microenvironment-mimicking systems are effective for cell-based screening tests of drug and chemical toxicity. Filamentous bacteriophages have diverse applications in material science, drug delivery, tissue engineering, energy, and biosensor development. Specifically, genetically modified bacteriophages have the potential to deliver therapeutic molecules or genes to targeted tumor tissues. The engineered bacteriophages in this study significantly enhanced endothelial cell migration and tube formation within the extracellular matrix (ECM). Compared to TGF-β1 alone and non-modified phage, the presence of TGF-β1 on the bacteriophages demonstrated superior performance as a continuous stimulant in the microenvironment, effectively promoting these angiogenic processes. Assays, including RT-qPCR, ELISA, and fluorescence microscopy, confirmed the expression of angiogenic markers such as CD31, validating the formation of 3D angiogenic structures. Our findings indicate that TGF-β1 displayed by bacteriophages likely acted as a chemotactic factor, promoting the migration, proliferation, and tube formation of endothelial cells (ECs) within the ECM. Although direct contact between ECs and bacteriophages was not explicitly confirmed, the observed effects strongly suggest that TGF-β1-RGD bacteriophages contributed to the stimulation of angiogenic processes. The formation of angiogenic structures by ECs in the ECM was confirmed as three-dimensional and regulated by the surface treatment of microfluidic channels. These results suggest that biocompatible TGF-β1-displaying bacteriophages could continuously stimulate the microenvironment in vitro for angiogenesis models. Furthermore, we demonstrated that these functionalized bacteriophages have the potential to be utilized as versatile biomaterials in the field of biomedical engineering. Similar strategies could be applied to develop angiogenic matrices for tissue engineering in vitro assay.
3. Reviewer Comment:
"Amplification and purification of phage (Line 90): This section needs better in-depth explanation of PEG removal from phages during purification as it affects ECM formation!"
Response:
We agree with the reviewer’s suggestion. So, we change the text describing the purification of phage
From: The final soluble phage supernatant was dialyzed against MilliQ water overnight to remove the remaining PEG.
By:
[line 123] To ensure minimal PEG contamination, the final soluble phage solutions were dialyzed against MilliQ water overnight at 4°C using a D-Tube Dialyzer Maxi (Millipore, USA) with a 6-8 kDa molecular weight cutoff [29].
[29] Yamamoto, K.R., et al., Rapid bacteriophage sedimentation in the presence of polyethylene glycol and its application to large-scale virus purification. Virology, 1970. 40(3): p. 734-44.
4. Reviewer Comment:
"Enzyme-linked immunosorbent assay (ELISA): The control of non-modified phage f88 is crucial, but what about ECM-associated enzymes or crude enzymes/proteins following phage enrichment in hosts (endotoxin-associated)?"
Response:
We appreciate the reviewer's concern about endotoxin and ECM-related enzyme contamination. We did not employ endotoxin removal procedures in this study. However, we fully agree that it is very important to control endotoxin and bacterial enzyme contamination to ensure accurate results. In our future studies, we will include controls to monitor putative contamination by endotoxin and bacterial enzymes.
We have added the following text to the discussion:
[line 448] Additionally, Doub et al. have reported that the use of bacteriophages can lead to endotoxin contamination and the formation of bacterial biofilms [39, 40]. Thus, to better understand the efficacy of bacteriophages, it is crucial to implement controls for endotoxin and bacterial enzyme contamination in future studies.
39. Doub, J.B., Risk of Bacteriophage Therapeutics to Transfer Genetic Material and Contain Contaminants Beyond Endotoxins with Clinically Relevant Mitigation Strategies. Infect Drug Resist, 2021. 14: p. 5629-5637.
40. Luong, T., et al., Standardized bacteriophage purification for personalized phage therapy. Nat Protoc, 2020. 15(9): p. 2867-2890.
5. Reviewer Comment:
"Tube formation is not an ideal criterion for angiogenic stimulation, as even hypoxia or lack of oxygen can induce such stress and not just reported components!"
Response:
We agree with the reviewer comment that hypoxia or oxygen deprivation are common issues in 3D cell cultures, which can induce stress and impact experimental outcomes.
We have added the following text to the discussion:
[line 389] Thus, we utilized a lab-on-a-chip platform that facilitates convenient media exchange, allowing for better control of oxygen levels [33]. To overcome this problem, previous studies modifying the ECM, cells, and media have been reported [34, 35]. Furthermore, our results suggest that TGF-β1 displayed by bacteriophages functions as a chemotactic factor, promoting endothelial cell migration and angiogenesis. Although we did not directly observe a physical interaction between ECs and the bacteriophages, the increase in cell migration and tube formation observed is consistent with known roles of TGF-β1 in stimulating chemotactic responses, as evidenced by CD31 expression in ECs. This effect may further enhance cell adhesion and migration in the presence of RGD. We believe that our study is closely related to concepts modifying the ECM, and provides additional functionality as a support for cell culture. This study confirmed the safety and long-term effectiveness of the potential materials. We did not observe abnormalities in the cells. Bacteriophage is a material that proves bio-safety currently undergoing phase 2 clinical trials [36].
33. Quintard, C., et al., A microfluidic platform integrating functional vascularized organoids-on-chip. Nat Commun, 2024. 15(1): p. 1452.
34. Hussey, G.S., M.C. Cramer, and S.F. Badylak, Extracellular Matrix Bioscaffolds for Building Gastrointestinal Tissue. Cell Mol Gastroenterol Hepatol, 2018. 5(1): p. 1-13.
35. Abbott, A., Cell culture: biology's new dimension. Nature, 2003. 424(6951): p. 870-2.
36. National Institute of Allergy and Infectious Diseases (NIAID) A Phase 1b/2 Trial of the Safety and Microbiological Activity of Bacteriophage Therapy in Cystic Fibrosis Subjects Colonized with Pseudomonas Aeruginosa. [(accessed on 28 February 2023)]; Available online: https://clinicaltrials.gov/ct2/show/NCT05453578.
6. Reviewer Comment: "The results and discussion must be critically evaluated, supported by previous studies to make it better!"
Response: We thank the reviewer for this suggestion and have expanded the Results and Discussion sections as follows (also addressing some further reviewer comments).
New text added to the discussion:
[line 392] Furthemore, our results suggest that TGF-β1 displayed by bacteriophages serves as a chemotactic factor, enhancing endothelial cell migration and promoting angiogenesis. Although we did not directly observe a physical interaction between ECs and the bacteriophages, the increase in cell migration and tube formation aligns with TGF-β1's role in stimulating chemotactic responses, as evidenced by CD31 expression in ECs as previously described [37]. This effect could potentially enhance cell adhesion and migration, particularly in the presence of RGD. One of the most challenges in tissue engineering is achieving long-term cell culture viability in mimicking the oxygen and nutrient delivery systems found in vivo [33]. By utilizing genetically modified filamentous bacteriophages, we demonstrate a promising alternative that not only supports cell migration but also supporting growth factor in long-term cultures at 3D cell culture systems. The growth factor displayed bacteriophages enhance angiogenesis and endothelial cell migration, key components of vascularization. Compared to traditional cell culture, bacteriophages offer a level of bio-safety and adjustability that makes them ideal for use in scaffolds [38]. Despite these findings, the risk of endotoxin contamination must be carefully addressed in future studies, as highlighted by previous research [39, 40]. Effective control of endotoxin mechanisms will be crucial to ensure the safety and reproducibility of bacteriophage-based biomedical applications.
33. Quintard, C., et al., A microfluidic platform integrating functional vascularized organoids-on-chip. Nat Commun, 2024. 15(1): p. 1452.
37. Meixian, L., et al., Saponin â… from Shuitianqi () inhibits metastasis by negatively regulating the transforming growth factor-beta1/Smad7 network and epithelial-mesenchymal transition in the intrahepatic metastasis Bagg's Albino/c mouse model. J Tradit Chin Med, 2024. 44(4): p. 642-651.
38. Yoon, J., et al., Angiogenic Type I Collagen Extracellular Matrix Integrated with Recombinant Bacteriophages Displaying Vascular Endothelial Growth Factors. Adv Healthc Mater, 2016. 5(2): p. 205-12.
39. Doub, J.B., Risk of Bacteriophage Therapeutics to Transfer Genetic Material and Contain Contaminants Beyond Endotoxins with Clinically Relevant Mitigation Strategies. Infect Drug Resist, 2021. 14: p. 5629-5637.
40. Luong, T., et al., Standardized bacteriophage purification for personalized phage therapy. Nat Protoc, 2020. 15(9): p. 2867-2890.
7. Reviewer Comment:
"comments on the tables and figures:”
“The articles need a flow diagram or chart to explain the overall procedure from the production of phage/modified phage enrichment, its purification and protocol for effective generation of engineered scaffold, and its comparison with normal phages and PEG alone to demonstrate the effect is of phage or modified phage and not of any other external factors!”
Response:
We appreciate the reviewer’s suggestion to clarify the overall procedure through visual representation. To clarify this, we have revised Figure 1 to include a schematic diagram of the extracellular matrix (ECM), collagen type I, showing the interaction between phages and the ECM during scaffold generation.
Additionally, in the Materials and Methods section, we have clarified our process to ensure minimal PEG contamination. We have added the following statement:
[line 233]
Figure 1. Schematic illustration of construction of the functional filamentous bacteriophages and cell seeding and cell culturing in microfluidic chips. (a) The structure of functionalized filamentous bacteriophage genetically engineered site. (b) Dialysis for removal of contaminants and unbound PEG from bacteriophage elution solution. (c) Microfluidics chip of DAX-1. (d) Polymerization of the type I hydrogel and bacteriophages at the gel channel. (e) Seeding of cells with medium at left medium channel. (f) Filling the medium at right medium channel. (g) Monitoring of HUVECs invasion during TGF-b1 with cells interaction in microfluidics chip.
[line 123] To ensure minimal PEG contamination, the final soluble phage solutions were dialyzed against MilliQ water overnight at 4°C using D-Tube Dialyzer Maxi (Millipore, USA) with a 6-8 kDa molecular weight cutoff [29].
[29] Yamamoto, K.R., et al., Rapid bacteriophage sedimentation in the presence of polyethylene glycol and its application to large-scale virus purification. Virology, 1970. 40(3): p. 734-44.
8. Reviewer Comment:
"The references are appropriate but the recent articles regarding the scaffolds can improve the overall quality of the literature review along with phage-related recent references as mentioned above.”
References:
Sanmukh, S. G., Dos Santos, N. J., Barquilha, C. N., De Carvalho, M., Dos Reis, P. P., Delella, F. K., Carvalho, H. F., Latek, D., Fehér, T., Felisbino, S. L."Bacterial RNA virus MS2 exposure increases the expression of cancer progression genes in the LNCaP prostate cancer cell line". Oncology Letters 25, no. 2 (2023): 86. https://doi.org/10.3892/ol.2023.13672
Turnbull, G., Clarke, J., Picard, F., Riches, P., Jia, L., Han, F., Li, B., & Shu, W. (2017). 3D bioactive composite scaffolds for bone tissue engineering. Bioactive materials, 3(3), 278–314. https://doi-org.proxy.insermbiblio.inist.fr/10.1016/j.bioactmat.2017.10.001”
Response:
We appreciate the reviewer’s suggestion to enhance the quality of the literature review by incorporating more recent articles on scaffolds and phages.
We add the following updated references about tissue engineering and cancer therapy by using filamentous bacteriophages at the Introduction.
New:
[line 56] Recently, scaffolds with biological activity, such as bone morphogenic proteins (BMPs), have been utilized in tissue engineering. Additionally, various materials, including polymers, hydrogels, metals, ceramics, and bio-glass, are being explored for their potential applications in this field [6].
6 Turnbull, G., Clarke, J., Picard, F., Riches, P., Jia, L., Han, F., Li, B., & Shu, W. (2017). 3D bioactive composite scaffolds for bone tissue engineering. Bioactive materials, 3(3), 278–314. https://doi-org.proxy.insermbiblio.inist.fr/10.1016/j.bioactmat.2017.10.001”
Add:
[line 84] Bacteriophages are rapidly being developed and utilized as functional biomaterials in various fields, including SARS-CoV-2 vaccines [14], molecularly imprinted polymers [15], lithium-ion battery [16], SPR electrochemical biosensors [17], polyethyleneimine drug carriers [18], monoclonal antibody production (phage display libraries) [19], TLR9 signal pathway cancer therapy [20] and carbon nanofiber [21].
14. Staquicini, D.I., et al., Design and proof of concept for targeted phage-based COVID-19 vaccination strategies with a streamlined cold-free supply chain. Proc Natl Acad Sci U S A, 2021. 118(30).
15. Baek, I.H., et al., Detection of Acidic Pharmaceutical Compounds Using Virus-Based Molecularly Imprinted Polymers. Polymers (Basel), 2018. 10(9).
16. Winton, A.J. and M.A. Allen, Rational Design of a Bifunctional Peptide Exhibiting Lithium Titanate Oxide and Carbon Nanotube Affinities for Lithium-Ion Battery Applications. ACS Appl Mater Interfaces, 2023. 15(6): p. 8579-8589.
17. Xu, P., et al., Screening of specific binding peptides using phage-display techniques and their biosensing applications. TrAC Trends in Analytical Chemistry, 2021. 137(116229).
18. Dong, X., et al., Hybrid M13 bacteriophage-based vaccine platform for personalized cancer immunotherapy. Biomaterials, 2022. 289: p. 121763.
19. Mairaville, C., et al., Identification of monoclonal antibodies from naive antibody phage-display libraries for protein detection in formalin-fixed paraffin-embedded tissues. J Immunol Methods, 2024. 532: p. 113730.
20. Sanmukh, S.G., et al., Bacterial RNA virus MS2 exposure increases the expression of cancer progression genes in the LNCaP prostate cancer cell line. Oncol Lett, 2023. 25(2): p. 86.
21. Szot-Karpinska, K., et al., Modified Filamentous Bacteriophage as a Scaffold for Carbon Nanofiber. Bioconjug Chem, 2016. 27(12): p. 2900-2910.

Round 2
Reviewer 1 Report
Comments and Suggestions for Authors
Dear Authors,
thanks for having answered to my concerns.
Just one has been misinterpreted: I intende to say that,. besides Figure 6, also all the other Figures and Tables, where necessary, should be completed with an appropriate statistical analysis.
Comments on the Quality of English LanguageIt should be double checked.
Author Response
Thank you for your valuable feedback and for pointing out the need for statistical analysis across all figures and tables. I apologize for any confusion in my previous response.
You are absolutely correct that, in addition to Figure 6, all other figures and tables should include appropriate statistical analysis, where necessary. All experiments were performed in triplicate; however, only the data in Figure 6 yielded statistically significant results. For the remaining figures and tables, while statistical tests were conducted, the results did not meet the threshold for statistical significance (p-value ≥ 0.05). Therefore, we have chosen to present these data as trends, reflecting the observed experimental outcomes without claiming statistical significance.
In the Materials and Methods section, we have highlighted the following text in yellow and updated it accordingly:
While all experiments were performed in triplicate, only the data presented in Figure 6 yielded statistically significant results. For other figures and tables, statistical analyses were conducted; however, the results did not pass the significance threshold (p-value ≥ 0.05). Consequently, these data are presented as trends, reflecting the experimental outcomes as observed without statistical significance.
Reviewer 2 Report
Comments and Suggestions for Authors
The article after revision is significantly improved but it will need minor changes in figure 6 (e) & 6 (f) as the lebals to the bar are confusing and must be corrected before publication.
Comments on the Quality of English LanguageThe article looks fine
Author Response
We sincerely appreciate the reviewer’s insightful suggestions and positive feedback on our revised manuscript. In particular, we thank the reviewer for highlighting the issue with the labels in Figures 6(e) and 6(f). We have addressed this by correcting the confusing labels in both figures to ensure clarity. In addition, we have modified the figure 6 according to the following statement: